# The Effect of Non-Invasive Brain Stimulation (NIBS) on Attention and Memory Function in Stroke Rehabilitation Patients: A Systematic Review and Meta-Analysis

**DOI:** 10.3390/diagnostics11020227

**Published:** 2021-02-03

**Authors:** Takatoshi Hara, Aturan Shanmugalingam, Amanda McIntyre, Amer M. Burhan

**Affiliations:** 1Department of Rehabilitation Medicine, School of Medicine, Jikei University, Tokyo 105-8461, Japan; t_hara1019@jikei.ac.jp; 2St. Joseph’s Health Care, Parkwood Institute Mental Health, London, ON N6A 4V2, Canada; Aturan.Shanmugalingam@sjhc.london.on.ca; 3Parkwood Institute Research, Parkwood Institute, London, ON N6C 0A7, Canada; Amanda.McIntyre@sjhc.london.on.ca; 4Department of Psychiatry and Medicine, Schulich School of Medicine and Dentistry, Western University, London, ON N6A 5C1, Canada; 5Ontario Shores Centre for Mental Health Sciences, 700 Gordon Street, Whitby, ON L1N 5S9, Canada; 6Department of Psychiatry, University of Toronto, Toronto, ON M5T 1R8, Canada

**Keywords:** stroke, non-invasive brain stimulation, transcranial magnetic stimulation, transcranial direct current stimulation

## Abstract

Background: In recent years, the potential of non-invasive brain stimulation (NIBS) for therapeutic effects on cognitive functions has been explored for populations with stroke. There are various NIBS methods depending on the stimulation site and stimulation parameters. However, there is no systematic NIBS review of post-stroke cognitive impairment with a focus on stimulation sites and stimulation parameters. The purpose of this study is to conduct a systematic review and meta-analysis on effectiveness and safety of NIBS for cognitive impairment after a stroke to obtain new insights. This study was prospectively registered with the PROSPERO database of systematic reviews (CRD42020183298). Methods: All English articles from MEDLINE, Scopus, CINAHL, Embase, PsycINFO, and CENTRAL were searched from inception up to 31 December 2020. Randomized and prospective controlled trials were included for the analysis. Studies with at least five individuals post-stroke, whereby at least five sessions of NIBS were provided and using standardized neuropsychological measurement of cognition, were included. We assessed the methodological quality of selected studies as described in the Physiotherapy Evidence Database (PEDro) scoring system. Results: A total of 10 studies met eligibility criteria. Six studies used repetitive transcranial magnetic stimulation (rTMS) and four studies used transcranial direct current stimulation (tDCS). The pooled sample size was 221 and 196 individuals who received rTMS and tDCS respectively. Eight studies combined general rehabilitation, cognitive training, or additional therapy with NIBS. In rTMS studies, target symptoms included global cognition (*n* = 4), attention (*n* = 3), memory (*n* = 4), working memory (WM) (*n* = 3), and executive function (*n* = 2). Five studies selected the left dorsolateral prefrontal cortex (DPLFC) as the stimulation target. One rTMS study selected the right DLPFC as the inhibitory stimulation target. Four of six studies showed significant improvement. In tDCS studies, target symptoms included global cognition (*n* = 2), attention (*n* = 4), memory (*n* = 2) and WM (*n* = 2). Three studies selected the frontal area as the stimulation target. All studies showed significant improvement. In the meta-analysis, rTMS showed a significant effect on attention, memory, WM and global cognition classified by neuropsychological tests. On the other hand, tDCS had no significant effect. Conclusions: In post-stroke patients with deficits in cognitive function, including attention, memory, and WM, NIBS shows promising positive effects. However, this effect is limited, suggesting that further studies are needed with more precision in stimulation sites and stimulation parameters. Future studies using advanced neurophysiological and neuroimaging tools to allow for a network-based approach to treat cognitive symptoms post-stroke with NIBS are warranted.

## 1. Introduction

After stroke, cognitive impairment may lead to significant functional impairment. The main cognitive symptoms are memory, attention, executive, and social behavior impairment [1]. In a national epidemiological cohort study after brain injury in the chronic phase, Nakajima et al. [1] reported that the most common cognitive symptoms were memory impairment (90%), attention disorder (82%), and executive function impairment (75%). Alas, cognitive dysfunction can present quite heterogeneously post-stroke. The scope and severity of cognitive symptoms depend on many factors, including injury mechanism in addition to demographic and social factors. Impairment of attention disorder is particularly common and can be seen in 24–51% of cases at time of discharge from acute care [2,3]. Cognitive impairment may persist beyond the acute phase of recovery; for example, memory impairment persists in 11–31% at one year post-stroke [4,5]. Further, these cognitive issues cause significant functional limitations, including impaired rehabilitation effort, impaired ability to resume work, and the need for additional support [6,7]. 

Cognitive rehabilitation is the mainstay of treatment for cognitive deficits associated with stroke [8]. Cognitive rehabilitation focuses on compensatory strategies to improve an individual’s functioning and facilitate learning. Unfortunately the evidence of this effect remains limited [9]. Several systematic reviews have been reported regarding rehabilitation of cognition post-stroke [10,11,12]. For example, Chung et al. concluded that there is insufficient high-quality evidence to confirm the effect of cognitive rehabilitation on executive function [10]. 

Recently, the role of non-invasive brain stimulation (NIBS) in rehabilitation of cognitive impairments post-stroke has attracted significant attention [13,14]. In general, NIBS techniques use electrical and/or magnetic energy to induce change in excitability of the underlying brain cortex in a non-invasive fashion and potentially induce long-lasting neuroplastic changes. rTMS and tDCS have different mechanisms of action on the cerebral cortex. TMS produces a time-varying magnetic field that flows perpendicular to the stimulating coil, which then induces electric currents that are generally parallel to the coil in the underlying cortical tissue. Different stimulation frequencies have different effects on the activity of the cerebral cortex, with high-frequency (>5 Hz) stimulation facilitating local neuronal excitability and low-frequency (<1 Hz) stimulation showing inhibitory effects [13,14]. On the other hand, tDCS is applied using a battery-powered direct current generator connected to two relatively large anodal and cathodal sponge-enclosed rubber electrodes positioned over the scalp. It is generally considered that anodal tDCS facilitates action potential of the underlying areas of the cortex, whereas cathodal tDCS inhibits the action potential [13,14]. Although there are several methods for NIBS, rTMS and tDCS are currently the mainstream stimulation methods in clinical applications [13,15,16]. Both rTMS and tDCS have been applied in the field of psychiatric disorders and especially to treat depression [17,18]. In recent years, the potential of NIBS to have therapeutic effects on cognitive function has been explored for stroke populations as well [19,20,21]. We have previously reported a case in which improvement of cognitive deficits was achieved by using rTMS combined with intensive rehabilitation following brain injury. Furthermore, the use of single photon emission computer tomography demonstrated changes in perfusion in the rTMS target sites and areas surrounding the targets [20]. Due to the heterogeneous nature of cognitive impairment after a stroke or brain injury, it is difficult to determine specific stimulation sites, stimulation parameters, and stimulation durations for cognitive rehabilitation. In fact, in our case report, different stimulation sites and parameters were individually selected from images and symptoms before stimulation [20]. Currently, there is no systematic review of the effects of NIBS on cognitive impairment after stroke. Therefore, we aim to conduct a systematic review and meta-analysis on the effectiveness and safety of NIBS for cognitive impairment after stroke to obtain new insights into its utility in this population. Given that NIBS could be a promising complementary treatment when used in combination with conventional cognitive rehabilitation, we also aim to examine the relationship between NIBS and cognitive rehabilitation, including stimulation site, stimulation parameters, neuropsychological tests, and secondary outcomes used.

## 2. Materials and Methods

This study was prospectively registered with the PROSPERO database of systematic reviews (CRD42020183298).

### 2.1. Literature Search Strategy

The following sources were searched from inception and up to 31 December 2020 for literature published in the English language: MEDLINE, Scopus, CINAHL, Embase, PsycINFO, and CENTRAL. Selected keywords included Stroke, Cerebral Vascular Accident, Ischemic Stroke, Hemorrhagic Stroke, Non-invasive brain stimulation, Transcranial magnetic stimulation, Theta-burst stimulation, Quadripluse stimulation, Transcranial Electrical Stimulation, Transcranial direct-current stimulation, Transcranial Alternating current stimulation, Cognition, Memory, Attention, Executive functioning. Unilateral spatial neglect (USN) was excluded in this review because USN has reported several independent reviews with robust evidence that inhibitory stimulation to the left posterior parietal cortex (PPC) is effective for the stimulation site [22]. Variations of keywords were individualized for each scientific database. All retrieved articles were reviewed to ensure relevant articles were included for data synthesis. The PubMed search strategy is illustrated in Appendix A.

### 2.2. Study Selection

Articles reporting on randomized and prospective controlled trials (RCT and PCT, respectively) were included for review. We included studies in which NIBS was used for cognitive rehabilitation or training post-stroke, reported cognitive function pre- and post-intervention, and included a minimum of five sessions of NIBS. Articles reporting on protocols, in-progress trials, retrospective studies, or case reports were excluded. We included studies reporting on at least five patients, who were 18–85 years old post-stroke. Two authors (TH and AS) independently reviewed all potential studies for inclusion against the eligibility criteria. They examined the title and abstract and, where necessary, the full text of studies to assess if they were eligible for inclusion. If they could not reach agreement by discussion, a third author (AB) made the final decision about eligibility.

### 2.3. Data Extraction and Synthesis

Two authors (TH and AS) independently used a standard form to extract study characteristics and outcome data from the studies. Discrepancies were checked against the original data. A third author (AB) made the final decision in the case of disagreement. Data extracted from each study included author, year, sample size, sex, age, time between onset and treatment, target symptom, stimulation site, each NIBS parameter, rehabilitation, outcome measures, and results. Several studies evaluated symptoms of cognitive impairment after stroke (attention, working memory (WM), memory, cognition and executive) using neuropsychological tests. We categorized by symptoms at the data extraction stage. A meta-analysis was performed on attention, WM, memory, and cognition. Because rTMS and tDCS have different mechanisms of stimulation on the cerebral cortex, they were performed separately. Regarding tDCS, attention and memory were analyzed by dividing into two components. We thought new insights could be obtained by interpreting the relationship between the stimulation site, stimulation parameters, and the results of neuropsychological tests. As such, we decided to make this the focus of our study.

### 2.4. Methodological Quality

We assessed the methodological quality of selected studies as described in the Physiotherapy Evidence Database (PEDro) scoring system [23]. This assessment has 11 items on study quality that are answered with yes (score = 1) or no (score = 0). The first item is a measure of external validity and is not used in calculating the final score. Based on this assessment, all studies were given a level of evidence (LoE) according to a modified Sackett Scale [24]. PEDro is widely used in systematic reviews in the rehabilitation area, and the PEDro tool has been used to score over 46,000 RCTs across 14 physiotherapy areas including neurorehabilitation. In NIBS for stroke, some clinical trials have been conducted on a wide range of symptoms, such as upper and lower limbs, aphasia, and spasticity. Additionally, some techniques combining rehabilitation have also been tried [15,19,20,21,25]. From this background, NIBS for cognitive impairment is also considered as one of the complimentary rehabilitation methods. Therefore, we adopted PEDro, which is typically used for systematic reviews in rehabilitation fields. 

### 2.5. Statistical Analysis

We decided to perform a meta-analysis from the extracted studies. Meta-analyses have the benefit of assessing the strength of evidence from a treatment from multiple studies; the aim is to determine if there is an effect, whether positive or negative, and to obtain a single summary estimate of the effect, as opposed to singe estimates from individual studies. In meta-analysis, for each outcome related to continuous data, we calculated a pooled estimate of the standardized mean difference (SMD) with 95% confidence interval (CI) between pre- and post-intervention. We used a generic inverse variance method and random effects model to combine individual results. In the general inverse variance method, each study is given a weight which is the inverse of the variance of the effect estimate (i.e., one over the square of its standard error). For larger studies, which have smaller standard errors, they are given more weight than smaller studies, which have larger standard errors. This choice of weight minimizes the imprecision (uncertainty) of the pooled effect estimate. Sample size aside, it is generally unlikely that all studies are functionally equivalent. Given the potential heterogeneity between studies, we chose to use a random effects model which would assign different weights to studies, in contrast to a fixed effects model which calculates a weighted average. The threshold for significance was set at *p* < 0.05. For all statistical comparisons we used Review Manager Software Version 5.3 from the Cochrane Collaboration. We used the I^2^ statistic to assess heterogeneity. I^2^ > 50% was considered to reflect substantial heterogeneity [26]. We were unable to perform the funnel plot method for assessment of reporting biases given that this can be done only when there were at least 10 studies included in the meta-analysis [27]. Statistical analysis was conducted by each neuropsychological test and NIBS.

## 3. Results

### 3.1. Study Selection

We identified 540 records through the searches after removal of duplicates. No additional records from other sources were identified.

After screening the titles and abstracts, we excluded 527 records mainly because the studies were animal studies, abstracts only, articles reporting on protocols, in-progress trials, retrospective studies or case reports, systematic review, non-English language publications, and completely irrelevant articles. After further assessment, 13 studies were considered to meet the review inclusion criteria (Figure 1). Seven studies used rTMS (included intermittent theta-burst stimulation (iTBS)) and six studies used tDCS. Among the excluded articles, 2 had less than 5 sessions or 5 days of NIBS [28,29]. Not only did this not adhere to the recommended schedule for treatment of depression, which has been highly evidenced in recent years, but it is also suggested that long-term cognitive improvement is likely related to the number of stimulation session/days, with more stimulation sessions resulting in a longer-lasting response [30]. In addition, one study evaluated a functional independence measure with the stimulation site being the primary motor cortex. Therefore, this study was excluded [31]. 

### 3.2. Study Characteristics

The details of each study are provided in Table 1. In rTMS, the pooled sample size was 221 individuals who received rTMS with a sample size varying from 6 to 29 subjects per group. In terms of study design, all articles in this review were RCTs. The age range of the intervention group was 42.5–68.3 years, and for the control group, 47.3–66.8 years. The time between onset and treatment ranged from 19.1 days to 38 months. In tDCS, the pooled sample size was 196 individuals who received tDCS with a sample size varying from 5 to 25 subjects per group. In terms of study design, all articles in this review were RCTs. The age range of the intervention group was 54.5–65.3 years, and for the control group, 53.1–68.5 years. The time between onset of symptoms and treatment ranged from 26.2 days to 16.6 months (one study was unclear).

Nine studies were ranked as Level 1 evidence and one study as Level 2 evidence. In all the studies, subjects were randomly allocated to groups appropriately. With the exception of two studies, intervention and control groups were similar at baseline regarding the most important prognostic indicators. Blinding was highly variable among studies. All studies yielded at least one important outcome measure from more than 85% of the subjects initially assigned to a group. In addition, the results of statistical comparisons between groups and the presentation of point measures and measures of variability were adequately performed in many studies (70%, 60%).

The treatment characteristics, outcomes, and results for each study are listed in Table 2. Eight studies combined general rehabilitation, cognitive training, or additional therapy with NIBS. Of these, all patients received rehabilitation regardless of the intervention group or control group. In rTMS, five studies combined cognitive rehabilitation and cognitive training [32,33,34,35,36]. Yin et al. [34] and Lu et al. [35] combined computer-assisted cognitive training in all patients [34,35]. In tDCS, Shaker et al. [37] and Yun et al. [38] combined general rehabilitation with cognitive rehabilitation [37,38]. On the other hand, Park et al. [39] combined computer-assisted cognitive rehabilitation. Regarding assessment of cognitive impairment, six articles reported on cognition and memory [39] seven articles reported on attention, five articles reported on WM, and two articles reported on executive function. In terms of the stimulation pattern, four rTMS studies used excitability stimulation pattern. Tsai et al. [40] used 5 Hz rTMS and iTBS [40]. Kim et al. [36] used 1 Hz and 10 Hz of rTMS [36]. All articles involving tDCS used anodal simulation. Hosseinzadeh et al. [41] used anodal tDCS for the left superior temporal gyrus (STG) and cathodal tDCS for the right posterior parietal cortex (PPC) [41]. 

#### 3.2.1. Effect of rTMS

Figure 2 shows the relationship between the stimulation site and the neuropsychological test results herein. Four out of six studies using rTMS reported on cognition and memory as an outcome. Three studies reported on attention and WM. Two studies reported on executive function. Liu et al. [32] reported that 10 Hz stimulation applied to the left DLPFC had significant improvement which was observed in all of the neuropsychological test categories compared to the control group [32]. Li et al. [33] applied 5 Hz stimulation on the left DLPFC with cognitive improvements observed both in the intervention and control groups, while the rTMS group got more significant improvement [33]. In addition, they examined Resting-State functional Magnetic Resonance Imaging (rs-fMRI) before and after the intervention. They showed the difference in fractional amplitude of low-frequency fluctuation (fALFF) between intervention and control group; compared to the control group, the intervention group got higher fALFF values around the stimulation area. Tsai et al. [40] used 5 Hz stimulation or iTBS on the left DLPFC. The 5 Hz rTMS group showed significantly greater improvement than the sham group in RBANS total score, attention and delayed memory [40]. The iTBS group showed significantly greater improvement than the sham group in RBANS total score and delayed memory. The 5 Hz rTMS group exhibited a superior modulating effect in attention compared to the iTBS group. Yin et al. [34] reported that 10 Hz stimulation applied to the left DLPFC resulted in improved scores on the Montreal Cognitive Assessment (MoCA), Rivermead Behavior Memory Test (RBMT), and VST-B and -C compared to the control group [34]. They also examined rs-fMRI before and after the intervention in both groups. With 10 Hz left DLPFC rTMS treatments, ALFF in the left medial prefrontal cortex (PFC) was significantly increased. Furthermore, functional connectivity was noted to be significantly increased in the right medial PFC and the right ventral anterior cingulate cortex. Reported changes in MoCA and VST-C scores were significantly correlated with changes in ALFF and FC. Kim et al. [36] used 1 Hz (inhibitory) or 10 Hz (excitatory) stimulation on the left DLPFC. Neither groups showed improvement in any of the neuropsychological tests for attention, memory, WM, and executive function. However, mood state was noted to have significantly improved with 10 Hz stimulation [36]. Lu et al. [35] reported that 1 Hz stimulation applied to the right DLPFC yielded no significant difference on the MoCA, and the Loewenstein Occupational Therapy of Cognitive Assessment (LOTCA) between the intervention and control groups [35]. However, RBMT scores were higher in the intervention group. Two months after treatment, RMBT in the intervention group were higher than in the control group, but not MoCA and LOTCA scores. In the latter study, changes in brain-derived neurotropic factor (BDNF) were examined; BDNF decreased in the intervention group but it increased in the sham group. This change did not correlate with improvements in memory and general cognitive function. The meta-analysis of each symptom is shown in Figure 3.

Meta-analysis of attention included three interventions, with rTMS being associated with a significant improvement (SMD = 1.40, 95%CI, 0.36–2.44, *p* < 0.05). However, there was statistically significant heterogeneity between the trials (I^2^ = 83%). Meta-analysis of WM included three interventions; rTMS was again associated with a significant improvement (SMD = 1.35, 95%CI, 0.95–1.76, *p* < 0.05). There was no statistically significant heterogeneity between these trials (I^2^ = 0%). Meta-analysis of memory included three interventions, which also showed that rTMS was associated with a significant improvement (SMD= 0.76, 95%CI, 0.17–1.35, *p* < 0.05). However, there was statistically significant heterogeneity between the trials (I^2^ = 51%). Meta-analysis of cognition included three interventions and demonstrated that rTMS was associated with a significant improvement (SMD= 1.31, 95%CI, 0.87–1.75, *p* < 0.05). There was no statistically significant heterogeneity between the trials (I^2^ = 28%).

#### 3.2.2. Effect of tDCS

Figure 2 shows a summary of the relationship between the stimulation site and the neuropsychological test results described herein. Regarding tDCS, four studies assessed attention as the outcome, two assessed cognition, memory, and WM. Two studies used bilateral DLPFC and the bilateral prefrontal cortex (PFC) and one study used the fronto-temporal lobe (right or left). Shaker et al. [37] reported significant improvements in executive functions, including attention and memory, and cognitive functions [37]. In a study targeting bilateral PFC, Park et al. [39] reported that Digit Span, Visual Span, and MMSE were not significantly improved compared to the control group. However, the change ratio in auditory and the visual continuous performance test that relates to sustained and selective attention was significant in the intervention group [39]. Yun et al. [38] performed anodal tDCS on the left and right fronto-temporal lobe compared with sham stimulation. They observed improvement in digit span, visual span, and the verbal learning test in the left fronto-temporal stimulation and in the verbal learning test from right fronto-temporal stimulation; however, only the verbal learning test for both intervention groups was significantly improved compared with the control group [38]. Hosseinzadeh et al. [41] examined changes in attention function and found that anodal and cathodal tDCS groups showed improvement after 1 month and 3 months compared with baseline, but there was no significant difference between all groups [41]. The meta-analysis of each symptom is shown in Figure 4. Meta-analysis of visual and auditory attention included three interventions; notably tDCS was associated with no significant improvement (SMD = 0.06, 95%CI, −0.41–0.53, *p* > 0.05, SMD= 0.15, 95%CI, −0.35–0.65, *p* > 0.05). Meta-analysis of WM included three interventions and also noted tDCS was associated with no significant improvement (SMD = 0.23, 95%CI, −0.23–0.70, *p* > 0.05). Meta-analysis of verbal and visual memory included two interventions, which showed that tDCS was associated with no significant improvement (SMD = 0.28, 95%CI, −0.35–0.91, *p* > 0.05, SMD = 0.15, 95%CI, −0.36–0.66, *p* > 0.05). Meta-analysis of cognition included three interventions; tDCS was associated with no significant improvement (SMD = 0.14, 95%CI, −0.32–0.61, *p* > 0.05). There was no statistically significant heterogeneity between the trials in any of these meta-analyses.

#### 3.2.3. Safety

Among 10 included studies, eight reported no obvious side-effects. Two studies have reported minor adverse effects. Lu et al. [35] reported that one patient experienced transient headaches and dizziness in the intervention group [35]. Park et al. [39] reported that some patients had a prickling sensation (unknown number) at the sites of stimulation after tDCS [39].

## 4. Discussion

We performed a systematic review of the effect of NIBS on cognitive impairment for post-stroke populations. The results demonstrate evidence of positive effects on cognitive functioning, including attention, WM, and memory. However, at present, the number of studies are only 6 for rTMS and 4 for tDCs, illustrating the need for further research In terms of stimulation target and stimulation parameters, there is still limited evidence with significant variability.

In terms of stimulation sites, DLPFC was selected in all rTMS studies. Of the six, five selected excitatory stimulation for the left DLPFC (one study selected both HFS and LFS). Lu et al. [35] selected inhibitory stimulation for right DLPFC [35]. This is based on the inference that promotes activation of the left DPLFC from the theory of interhemispheric inhibition [42]. The reason for choosing DLPFC was that all studies indicated that DLPFC was an important site in cognitive function. Some studies have shown that DLPFC is associated with WM [32,36,40], is an important part of the Default mode network [33], is a hub of attentional function [32], and plays an important part in the central executive network (CEN), which is responsible for high-level cognitive functions such as control of attention and WM [34]. Except for the study by Hosseinzadeh et al. [41], excitatory stimulation of the frontal lobe was selected in tDCS studies [37,38,39]. It was stated that these frontal lobes were selected as sites with important roles in attention function, WM, and CEN [37,38,39]. In addition, some studies have stated that they have been selected from previous studies of NIBS for cognitive impairment in Parkinson’s disease (PD), Alzheimer’s disease (AD), and healthy individuals [38,39]. In fact, there are published reviews on the effects of NIBS on cognitive impairment. According to systematic reviews on NIBS for AD, in subgroup analyses, high frequency rTMS stimulation (HFS) showed a significant mean effect size of 1.64 (*p* < 0.001 95% CI, 1.03–2.27) compared to low frequency rTMS stimulation (LFS) [43]. In a systematic review of the effects of rTMS on AD, Liao et al. [44] reported HFS for right or bilateral DLPFC significantly improved the cognition (SMD = 1.06 95%, CI, 0.47–1.66 *p* < 0.05) [44]. A review of rTMS on cognition in mild cognitive impairment (MCI) pointed out the possibility that HFS has a better effect than LFS, although the effect size in all seven included studies was small at SMD = 0.48 (*p* = 0.01 95%CI, 0.12–0.84) [45]. In NIBS for PD, Dinkelbach et al. [46] suggested that it was effective to select DLPFC as the stimulation site for both rTMS and tDCS, HFS was effective in rTMS, and Anodal tDCS was effective in tDCS [46]. Therefore, as the report of NIBS for cognitive impairment in AD and PD, these findings suggest that excitatory stimulation pattern, particularly in the frontal lobe, may be effective post-stroke. Many reviews indicate that the region and patterning of excitatory stimulation, particularly centered on the frontal lobe, may be the key to observing improvement in cognitive impairment. The frontal lobe and DLPFC are said to be involved in executive function, memory, WM, and attention [47,48,49,50]. Therefore, these findings suggest that an excitatory stimulation pattern, particularly in the frontal lobe, may be effective for improving cognitive impairment post-stroke, but the evidence is still limited. In other diseases, it has been reported that HFS in the bilateral frontal region may be effective. The multiple target method has been proposed as a possible effective approach [51,52,53]. For future consideration, we would recommend studying these new stimulation parameters and methods for NIBS for cognitive dysfunction after stroke.

In the meta-analysis, rTMS showed a significant effect on all trials classified by neuropsychological tests but high heterogeneity was observed in attention and memory. On the other hand, tDCS had no significant effect in any trial but did not have concerns with heterogeneity. The significant effect of rTMS may be due to the selection of DLPFC in all studies. Some of the neuropsychological tests included multiple overlapping elements in our symptom-based classification for stroke. In addition, some of the extracted studies were not selected for meta-analysis due to their poor quality. Therefore, although rTMS showed a significant effect, there is insufficient evidence in terms of the number of studies and heterogeneity. In the future, we need further studies to clarify optimal stimulation site, stimulation parameters, intervention time from onset, and number of sessions.

This review found that few studies reported minor adverse events. The most concerning adverse event was a seizure after rTMS [15] and seizure and skin burn after tDCS [54]. No major adverse events were observed in the current review and no studies reported cognitive deterioration after NIBS. To establish routine use of NIBS for cognition post-stroke, it is necessary to establish a method for identifying the lowest-risk stimulation sites and stimulation parameters. For example, the navigation system can accurately identify the stimulation site and, in addition, reduce the propagation of the stimulation to the opposing brain function region. In particular, for patients with large brain lesions, precise setting of the stimulation site by the navigation system may be necessary [55]. LFS can reduce major adverse events such as seizure compared to HFS. Therefore, it is suggested that more clinical evidence is needed in the future regarding the relationship between safety and stimulation parameters to improve the effectiveness of treatment. To avoid severe side-effects when applying excitatory stimulation, it is necessary to consider not only the navigation system described above, but also the use of medication to reduce stimulation threshold, and monitoring of brain imaging via electroencephalogram.

In this systematic review, general or cognitive rehabilitation and supplementary cognitive training (included computer-assisted training) were conducted in five out of six rTMS studies and three out of four tDCS studies, of which three studies in both rTMS and tDCS showed improvement in cognitive outcomes. According to previous reports, NIBS in combination with rehabilitation has demonstrated significant improvements in physical functioning and aphasia after stroke [42,56]. Restoring impaired neural networks following brain injury is a viable means of promoting functional recovery. In such a situation, a strategy to promote network-related reorganization in the brain must be adopted [57]. NIBS may be a promising complementary treatment when used in conjunction with conventional therapies or training to enhance rehabilitation in patients with brain injury [19]. From the concept of rehabilitation aimed at improving neuroplasticity, NIBS combined with rehabilitation suggests the possibility of inducing a positive synergistic effect. In addition, this is thought to lead to not only modulation of neural connections, but also functional re-learning.

Based on this systematic review and our previous studies, to build evidence of NIBS for cognitive dysfunction after stroke, it is important not only to evaluate neuropsychological tests, but also to establish evidence for the effects of NIBS itself on neural networks. Regarding the effects of NIBS on neural networks, the mechanism of action differs between rTMS and tDCS, and the mechanism of action of NIBS itself still remains an important debate [16,17]. However, the potential for NIBS to have a positive impact on pathological rhythms in the network post-injury or by disease is consistent. Previous neuroimaging studies have reported that NIBS affects the cerebral cortex directly under the stimulation site or its functional-related brain regions based on neural networks [45,58,59].

In this systematic review, changes in brain activation were evaluated using neuroimaging and neuropsychological tests [33,34]. The common point in all is that a change in activity in the brain region was associated with the stimulation site. These results are consistent with previous NIBS studies for upper extremity and aphasia after stroke and TBI [42,60,61]. To make the clinical application of NIBS for cognitive impairment more robust, it is necessary to consider that the site of brain injury varies from patient to patient. The results obtained from NIBS may vary and would be reflected in changes in brain activity using neuroimaging along with neuropsychological tests. As indicated previously, NIBS affects not only the cerebral cortex under the stimulation site but also functional-related brain regions based on neural networks. For example, in a recent study of NIBS for aphasia, it was suggested that the stimulation site and parameter is selected depending on how the damaged language regions and homologous regions related to language acts on the recovery of language function. These selections are based on the duration of onset and the results of changes in brain activity by language tasks [42]. In terms of the relationship between NIBS and the effect of neural networks, Padmanbhan et al. [62] reported the relationship between brain function connectivity and post-lesion depression. Lesion locations associated with depression were highly heterogeneous and there were no consistent brain region related to depression [62]. Lesion locations were mapped to a connected brain circuit centered on the left DLPFC; the size of the damaged area alone could predict depression [62]. This same observation may be applied to the relationship between brain lesion and symptoms in cognitive impairment post-stroke. Kreuzer et al. [63] also described the relationship and neural connectivity between the DLPFC and anterior cingulate cortex in a review of NIBS for DLPFC. They suggested that rTMS for DLPFC has an effect on the anterior cingulate cortex, which is functionally related to DLPFC. Therefore, they argued that pre-clinical parameter studies combining TMS with neuroimaging are necessary [63]. Cognitive evaluation along with neuroimaging evaluation will lead to enhanced evidence of the effectiveness and accuracy of NIBS treatment, as well as the exploration of new insights and methods to manage cognitive impairment in post-stroke populations.

There were important limitations in this review to keep in mind. Firstly, this systematic review included studies that utilized various neuropsychological tests to evaluate the effects of NIBS. Some of the neuropsychological tests included multiple overlapping elements in our symptom-based classification for stroke. For example, TMA-A reflects attention, visual search, and working memory, while the TMT-B reflects executive processes such as cognitive set-shifting. In addition, neuropsychological tests performed at short intervention intervals can predispose to learning bias. Therefore, in NIBS for cognitive impairment for stroke, it is necessary to carefully consider the selection of neuropsychological tests.

Second, in the meta-analysis of this review, we were unable to perform sub-analyses to clarify different neuropsychological symptoms, stimulation site, and stimulation parameters due to the small number of studies extracted and variability in cognitive symptoms and stimulation location and parameters reported. Further studies are needed to clarify optimum location and stimulation parameters for the specific cognitive symptoms.

The third limitation was that in the extracted studies, the target patients were identified based on reported cognitive symptoms post-stroke as opposed to being classified based on brain imaging that establish the specific brain lesion. On the other hand, and from the neurorehabilitation standpoint, it is the cognitive symptoms and their functional impact that is considered more relevant. Functional neuroimaging and neurophysiological markers of cognitive and functional rehabilitation is likely needed to facilitate more precise application of NIBS in neurorehabilitation of cognitive impairment post stroke. Fourthly, in this study, we set 18–85 years old as inclusion criteria. As a result, the age ranges of both the intervention group and the control group were similar. However, age is an important factor influencing the effect of rehabilitation. Therefore, we had initially considered performing this sub-analysis. However, as a result of the lack of raw data by age bracket, this was not possible. In the future, there will be a need for age-classified sub-analysis.

Lastly, in addition to treating cognitive symptoms, it is important to evaluate the impact of NIBS on the recovery of activities of daily living as a more meaningful impact in post-stroke rehabilitation. Unfortunately, there were few such studies in this systematic review. Future studies need to evaluate changes in the activities of daily living to support the impact of NIBS as a complementary treatment combined with general and cognitive rehabilitation.

## 5. Conclusions

We performed a systematic review of the efficacy of NIBS on cognitive impairment post-stroke.

NIBS for cognitive functioning, including attention, memory, and WM in post-stroke, has been suggested to have a promising effect. However, the evidence for this effect remains limited, suggesting the need for further studies to address more precise application by improving stimulation sites and stimulation parameters. Excitatory stimulation of areas in the frontal lobe holds promise and needs to be explored further. Combining NIBS with neurorehabilitation to enhance the neuroplastic effect is also promising and needs further exploration. Finally, evaluation of brain activity at the stimulation site and related areas using advanced functional neuroimaging and neuropsychological tools would facilitate our understanding on the mechanism of action of NIBS on neural networks and would contribute to more precise neurorehabilitation targeting.

## Figures and Tables

**Figure 1 diagnostics-11-00227-f001:**
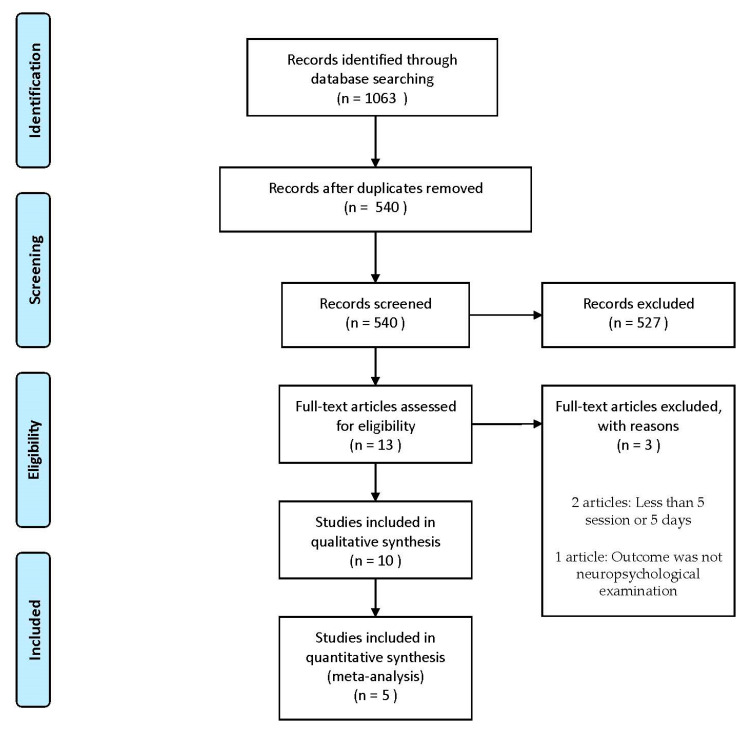
Study flow diagram.

**Figure 2 diagnostics-11-00227-f002:**
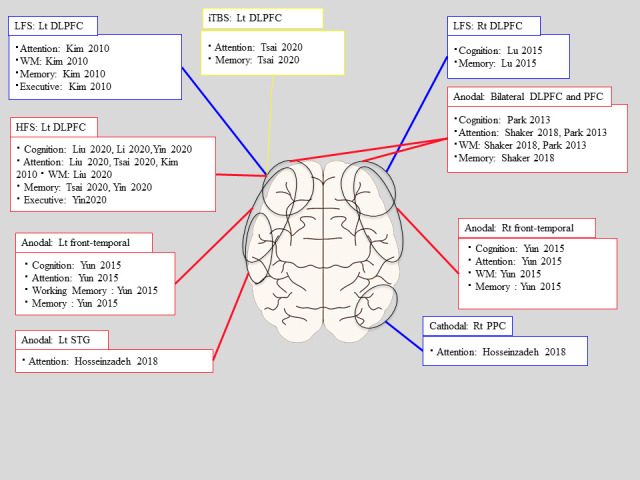
Relationship between stimulation site and neuropsychological test results in rTMS and tDCS. Abbreviations: DLPFC = Dorsolateral prefrontal cortex, PFC= Prefrontal cortex, PPC = Posterior parietal cortex, STG = superior temporal gyrus, LFS = Low frequency stimulation, HFS = High frequency stimulation, iTBS = intermittent theta-burst stimulation.

**Figure 3 diagnostics-11-00227-f003:**
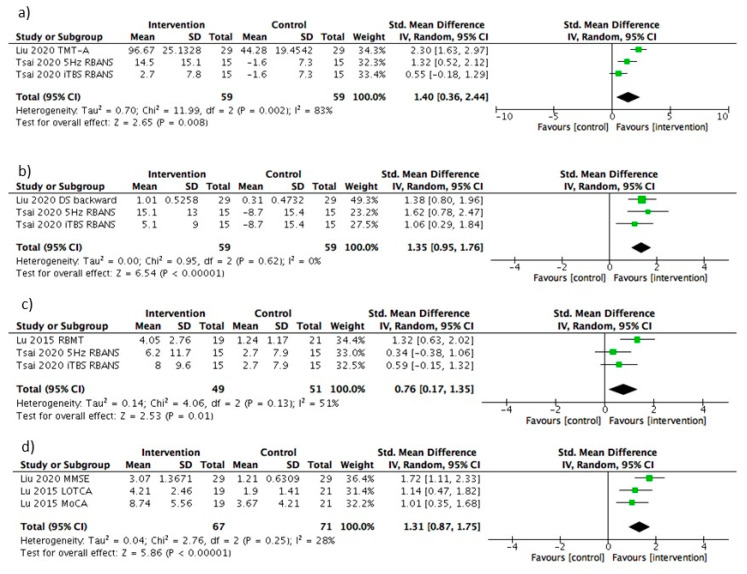
Forest plot of rTMS with neuropsychological tests outcome. (**a**) Attention, (**b**) Working memory, (**c**) Memory, (**d**) Cognition, Catherine Bergego Scale; CI, confidence interval; SD, standard deviation; Std, standard.

**Figure 4 diagnostics-11-00227-f004:**
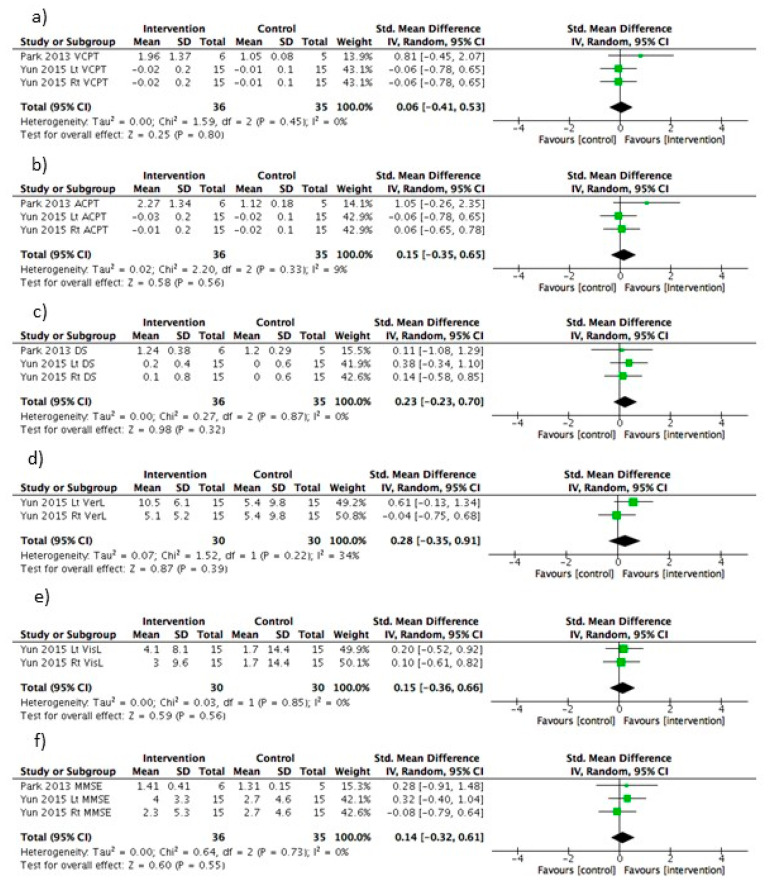
Forest plot of tDCS with neuropsychological tests outcome. (**a**) Attention in visual continuous performance test(VCPT), (**b**) Attention in auditory continuous performance test(ACPT), (**c**) Working Memory, (**d**) Memory in verbal learning test(VerL) (**e**) Memory in visual learning test(VisL), (**f**) Cognition, Catherine Bergego Scale; CI, confidence interval; SD, standard deviation; Std, standard.

**Table 1 diagnostics-11-00227-t001:** Study Characteristics.

**Study**	**Disease**	**Design—LoE**	**PEDro**	**Sample**	**Sex (M:F)**	**Age (SD)**	**Time between Stroke Onset and Treatment**
TMS							
Liu et al. 2020 [32]	Stroke	RCT-1 I + Re vs C(Sham) + Re	10	I: 29 C (Sham): 29	26:32	I: 58.5(6.24) C:57.7(7.25)	I: 8.79(1.84) C: 8.62(1.84) months
Li et al. 2020 [33]	Stroke	RCT-1 I + Re vs C(Sham) + Re	8	I: 15 C (Sham): 15	16:14	I: 65.5(3.68) C:64.5(4.72)	I: 22.73(8.05) C: 19.13(7.95) days
Tsai et al. 2020 [40]	Stroke	RCT-1 5 Hz rTMS vs iTBS vs C(Sham)	10	5 Hz rTMS: 11 iTBS: 15 C (Sham): 15	33:8	5 Hz rTMS:57.5(12.3) iTBS: 60.1(14.1) C:56.2(12)	5 Hz rTMS:33.3(26.4) iTBS: 18.5(20.2) C:38(7.9) months
Yin et al. 2020 [34]	Stroke	RCT-1 I + Re vs C(Sham) + Re	8	I: 16 C (Sham): 18	30:4	I: 56.7(12.9) C:58.2(11.3)	I: 52(38.25–98.75) C: 55(39.75–94.75) days
Lu et al. 2015 [35]	Stroke	RCT-1 I + Re vs C(Sham) + Re	8	I: 19 C (Sham): 21	25:15	I: 42.5(12.3) C:47.3(11.8)	61 (30–365) days
Kim et al. 2010 [36]	Stroke	RCT-1 1 Hz + Re vs 10 Hz + Re vs C(Sham) + Re	7	I: 12 (1 Hz 6, 10 Hz 6) C (Sham): 6	10:8	I: LFS 68.3(7.4) HFS: 53.5(16.9) C: 66.8(17.2)	I: LFS 404.4(71.7) HFS: 241.2(42.5) C: 69.7(39.0) days
tDCS							
Shaker et al. 2018 [37]	Stroke	RCT-1 I + Re vs C(Sham) + Re	7	I: 20 C (Sham): 20	40:0	I: 54.45(4.68) C: 53.05(6.32)	I: 14.05(1.53) C: 16.55(2.78) months
Hosseinzadeh et al. 2018 [41]	Stroke	RCT-1 Anodal vs Cathodal vs Sham vs Control(routine treatment)	6	Anodal: 25 Cathodal: 25 Sham: 25 Control: 25	49:51	Anodal: 58(8) Cathodal: 60(7) Sham: 59(7) Control: 59(8)	25–180 days
Yun et al. 2015 [38]	Stroke	RCT-2 Left;Re vs Right + Re vs Sham + Re	5	I:30(Left 15, Right15) C (Sham): 15	20:25	I: Left 60.9(12.9) I: Right58.9(15.0) C: 68.5(14.6)	I: Left 42.2(31.9) I: Right 38.1(27.0) C: 39.5(29.6) days
Park et al. 2013 [39]	Stroke	RCT-1 I + Re vs C(Sham) + Re	6	I: 6 C (Sham): 5	5:6	I: 65.3(14.3) C: 66.0(10.8)	I: 29.0(18.7) C:25.2(17.5) days

C = control group, HFS = High frequency stimulation, I = Intervention group, LFS = Low frequency stimulation, PEDro= Physiotherapy Evidence Database, RCT = Randomized controlled trials, Re = Rehabilitation.

**Table 2 diagnostics-11-00227-t002:** Individual Study Treatment Characteristics, Assessments, and Outcomes.

**Study**	**Disease**	**Targets**	**Stimulation Site**	**Parameter**	**Session**	**Rehabilitation**	**Assessments & Follow-Up**	**Results**
TMS								
Liu et al. 2020 [32]	Stroke	Cognition Attention WM	Left DLPFC	10 Hz 90%MT 700 pulses/session	20	Both groups of patients were given comprehensive cognitive function training. The cognitive function training was carried out on a touch screen computer.	TMT-A, DST, DS, MMSE, FIM	Intervention group was significantly improved in all assessment categories compared with the control.
Li et al. 2020 [33]	Stroke	Cognition	Left DLPFC	5 Hz 100%MT 2000 pulses/session	15	Routine cognitive training (included memory, attention, orientation, visual and spatial, judging and reasoning, executive capability) for 30 min/time, 1time/day and 5 days/week for total of 15 times in 3 weeks	MMSE, MoCA	Cognitive improvements were observed both in the intervention group and the control group, while the rTMS group got more significant improvement than the control group.
Tsai et al. 2020 [40]	Stroke	Attention, WM, Memory	Left DLPFC	rTMS:5 Hz 80%MT 600 pulses/session iTBS:3 pulses of 50 Hz repeated at 5 Hz for total 190 sec(600pulses)	10	None	RBANS, BDI	The 5 Hz rTMS group showed significantly greater improvement than the sham group in RBANS total score, attention, and delayed memory. The iTBS group showed significantly greater improvement than the sham group in RBANS total score and delayed memory. The 5 Hz rTMS group exhibited a superior modulating effect in attention compared to the iTBS group.
Yin et al. 2020 [34]	Stroke	Cognition, Memory, Executive	Left DLPFC	10 Hz 80%MT 2000 pulses/session	20	30-min computer-assisted cognitive rehabilitation referring to attention, executive function, memory, calculation, language and visuospatial skills after treatment.	MoCA, VST(a colored dots trail (A), a neutral words trail (B), and an incongruent- colored words trail (C)), RBMT, BI	The MoCA score in both groups increased significantly after four weeks and the score for the intervention group was significantly higher than that in the control group after treatments. The improvement of the RBMT score for the intervention group was significantly higher than that in the control group after treatments. The improvement of VST-B and -C for the intervention group was significantly higher than that in the control group after treatments.
Lu et al. 2015 [35]	Stroke	Cognition, Memory	Right DLPFC	1 Hz 100%MT 600 pulses/session	20	All patients received regular computer-assisted cognitive training for 30 min every day.	MoCA, LOTCA, RBMT Follow-up at 3 days and 2 months	No difference was observed between the intervention group and the control group for MoCA, LOTCA, and RBMT. However, RBMT was better in the intervention group. Two months after treatment, RMBT in the intervention group was higher than in the control group, but not MoCA and LOTCA scores.
Kim et al. 2010 [36]	Stroke	Attention, WM, Memory, Executive	Left DLPFC	1 Hz 900 pulses/ 10 Hz 450 pulses 80% MT	10	All patients received conventional cognitive rehabilitation two or three times a week for 2 wks.	DS, VS, VerL, VisL, VCPT, auditory CPT, a word-color test, ToL, BI, Beck Depression Inventory	There was no significant improvement about cognition in each intervention groups. However, mood state significantly improved with 10 Hz stimulation.
tDCS								
Shaker et al. 2018 [37]	Stroke	Attention, Memory	Bilateral DLPFC	2 mA × 30 min, The anode electrode was placed over the right and left DLPFC. The cathode was placed over the contralateral supraorbital area.	12	All patients received cognitive training program.	Computer-based cognitive therapy tool (attention and concentration, figural memory, reaction behavior, and logical reasoning.), FIM	There was a significant improvement in the scores of attention and concentration, figural memory, logical reasoning, reaction behaviour in both groups. However, the improvement was significantly higher in the intervention group compared to the control group.
Hosseinzadeh et al. 2018 [41]	Stroke	Attention	anodal: left STG, cathodal: Right PPC	2 mA/35 cm^2^ × 30 min	12	None	NIHSS, TMT, Beck test Follow-up at 1 and 3 months	In TMT, the control group, the Anodal group, and the Cathodal group showed improvement after 1 month and 3 months compared with baseline, but there was no significant difference between all groups. NIHSS, Beck test was improved in Anodal.
Yun et al. 2015 [38]	Stroke	Cognition, Attention, WM, Memory	fronto-temporal(T3 or T4)	2 mA/25 cm^2^ × 30 min The anodal stimulation was placed over T3 or T4.	15	All patients received cognitive rehabilitation.	MMSE, DS, VS, VerL, VisL, VCPT, ACPT, BI	Left anodal tDCS improved digit and visual span task and verbal memory. Right anodal tDCS improved only verbal memory between pre and post treatment. Left anodal tDCS significantly improved verbal memory compared to the other groups.
Park et al. 2013 [39]	Stroke	Cognition Attention, WM	Bilateral PFC	2 mA/25 cm^2^ × 30 min, The anodal stimulation was placed over bilateral PFC and the cathodal stimulation were placed over the non-dominant arm.	mean 18.5	All patients received computer assisted cognitive rehabilitation	DS, VS, CPT, MMSE	Intervention group was significantly improved in auditory and visual continuous performance compared with control.

DLPFC = Dorsolateral prefrontal cortex, PFC = Prefrontal cortex, PPC = Posterior parietal cortex, STG = Superior temporal gyrus, LFS = Low frequency stimulation, HFS = High frequency stimulation, iTBS= intermittent theta-burst stimulation, ACPT = Auditory continuous performance test, BDI = Beck’s Depression Inventory, BI = Barthel Index, CPT = continuous performance test, DS = Digit Span, DST= Digit Symbol Test, FIM = Functional Independence Measure, LOTCA = Loewenstein Occupational Therapy of Cognitive Assessment, MMSE = Mini-Mental State Examination, MoCA = Montreal Cognitive Assessment, MT = Motor threshold, NIHSS = National Institutes of Health Stroke Scale, RBMT = Rivermead Behavior Memory Test, TMT = Trail Making Test, ToL = Tower of London test, VCPT = visual continuous performance test, VerL = verbal learning test, VisL = visual learning test, VS = Visual span, VST = Victoria Stroop Test, WM = Working Memory.3.3. Outcomes.

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
