# Peer review of "The Effect of Non-Invasive Brain Stimulation (NIBS) on Attention and Memory Function in Stroke Rehabilitation Patients: A Systematic Review and Meta-Analysis"

_diagnostics, 2021, doi:10.3390/diagnostics11020227_

Round 1
Reviewer 1 Report
- Meta-analysis term should be included in the title and in the abstract.
- “Therefore, we aim to conduct a systematic review on effectiveness and safety of NIBS for cognitive impairment after a stroke and to obtain new insights” And if proceed a meta-analysis?
- What about the protocol registration of the systematic review? This should be indicated at the beginning of Methods and in the abstract.
- The authors should include all the data bases search equations in the Appendix 1, in order to allow readers to replicate their results.
- “Articles reporting on randomized and prospective controlled trials (RCT and PCT, respectively) were included for review.” Why only these designs?
- “Two authors (TH and AS) independently used a standard form to extract study characteristics and outcome data from the studies” Which information was extracted by the authors? In other words, put that information in Data extraction, instead of Data synthesis.
- Why did you perform a meta-analysis? In other words, I would like to see in the text the rationale to do this.
- “We used a generic inverse variance method and random effects model to combine individual results.” Why those methods?
- Sensitivity analysis?
- Please, reference the three excluded paper after reading the full text (transparency issues).
- The reasons are missing in the Figure 1.
- The rest of the paper I think it is great.
Author Response
Thank you for your feedback and for the opportunity to response to the review. Please see below point-by-point responses with references to the revised manuscript.
- Meta-analysis term should be included in the title and in the abstract.
The title and introduction has been modified to specify that we performed a meta-analysis (Page 3, Lines 108).
2. What about the protocol registration of the systematic review? This should be indicated at the beginning of Methods and in the abstract.
The registration number has been specified in both the abstract (Page 1, Lines 35) and methods section (Page 3, Lines 116-117).
- The authors should include all the data bases search equations in the Appendix 1, in order to allow readers to replicate their results.
All database search equations have been provided in Appendix 1.
- Articles reporting on randomized and prospective controlled trials (RCT and PCT, respectively) were included for review.” Why only these designs?
In evaluating the hierarchy of research evidence health interventions, there is significant consensus that RCTs and PCT designs are effective, appropriate and feasible over all other research designs with the exception of the systematic review and multi-centered trials (Evans et al. 2013). RCTs and PCTs have strong methodological quality because the processes used during their conduct minimize risk of confounding factors and therefore, their findings are reliable (Evans et al. 2013). For this reason, we only selected the RCT and PCT study design for inclusion.
[Evans, D. (2013). Hierarchy of evidence: A framework for ranking evidence evaluating healthcare interventions. Journal of Clinical Nursing, 12, 77-84.]
- Two authors (TH and AS) independently used a standard form to extract study characteristics and outcome data from the studies” Which information was extracted by the authors? In other words, put that information in Data extraction, instead of Data synthesis.
Thank you for this suggestion. We have specified the data extracted from each study and placed in in the appropriate methods section (Page 4, Lines 143-153).
- Why did you perform a meta-analysis? In other words, I would like to see in the text the rationale to do this.
Thank you for this suggestion. We have added our justification for a meta-analysis in the methods section (Page 4, Lines 180-183).
- “We used a generic inverse variance method and random effects model to combine individual results.” Why those methods?
In the general inverse variance method, each study is given a weight which is the inverse of the variance of the effect estimate (i.e. one over the square of its standard error). For larger studies, which have smaller standard errors, they are given more weight than smaller studies, which have larger standard errors. This choice of weight minimizes the imprecision (uncertainty) of the pooled effect estimate. Sample size aside, it is generally unlikely that all studies are functionally equivalent. Given the potential heterogeneity between studies we chose to use a random effects model which would assign different weights to studies, in contrast to a fixed effects model which simply calculates a weighted average. These details have been added to the appropriate methods section (Page 3, Lines 186-193).
- Sensitivity analysis?
A funnel plot method was not conducted to assess studies for reporting bias because it is done only when there were at least 10 studies included in the meta-analysis. We stated that in the text (page 5, Lines 196-197)
- Please, reference the three excluded paper after reading the full text (transparency issues).
Reference numbers 28,29,31 were articles excluded for review (Page 5, 3.1.Study Selection, Lines 202-207). Citation numbers have been updated and corrected.
- The reasons are missing in the Figure 1.
Page 6, Figure 1 has been revised to include all pertinent information.
Best regards,
Amer M. Burhan (senior corresponding author) on behalf of co-authors
Reviewer 2 Report
Thanks for recommending me as a reviewer. The purpose of this systematic review study was to conduct a systematic review on effectiveness and safety of NIBS for cognitive impairment after a stroke and to obtain new insights. If the authors complete revisions, the quality of the study will be further improved.
1. The introduction was well written. The authors described non-invasive brain stimulation (NIBS) well in the "Introduction section". However, rTMS & tDCS are different stimulation methods and have different mechanisms.
Why did the authors analyze rTMS and tDCS together in this systematic review?
If the author explains the difference between rTMS and tDCS more specifically in the introductory section, it may help readers understand.
2. Subject's age is too broad in the study's inclusion criteria (18-85 years old). Age is an important influencing factor in rehabilitation. Therefore, the authors need to further analyze the effects of NIBS according to age (e.g. young, old vs. old) using meta sub-analysis.
3. Although the research topic is comprehensive, this study finally analyzed only 13 studies. It is necessary to describe the research search and selection process in more detail.
4. If the number of studies involved was small, authors may perform a network meta-analysis' to analyze more studies. Are there any specific reasons for not performing network meta-analysis'?
5. line 176: Does "Additional records identified through other sources (n = 0)" in Figure 1 mean the gray literature (ex. government reports, technical Report, thesis)? I don't think I need to include it in the figure 1. If necessary, the authors suggest that the results of searches in the gray literature be described in text.
6. line 203-217: The limitations of the study need to be described in more detail.
Author Response
Thank you for your feedback and for the opportunity to response to the review. Please see below point-by-point responses with references to the revised manuscript.
- Why did the authors analyze rTMS and tDCS together in this systematic review? If the author explains the difference between rTMS and tDCS more specifically in the introductory section, it may help readers understand.
There are two main methods in the clinical application of non-invasive brain stimulation (NIBS), rTMS and tDCS which perform with different mechanisms. Initially, we considered performing a systematic review of the two mechanisms separately. However, we ultimately decided to assess both within a single manuscript. There were few studies to review to begin with, and if we only assessed one mechanism, there would always be a need to research the effects of the other. This was largely a pragmatic decision driven the publication of other, similar systematic reviews examining NIBS. rTMS and tDCS are the most applied methods in the field of psychiatric disorders and especially in depression. For example, Chu et al. (Neurodegeneration, 2021) examined both mechanisms in their review within an Alzheimer’s population. Additional text has been added to the introduction for clarity (Page 2-3, Lines 84-93).
- Subject's age is too broad in the study's inclusion criteria (18-85 years old). Age is an important influencing factor in rehabilitation. Therefore, the authors need to further analyze the effects of NIBS according to age (e.g. young, old vs. old) using meta sub-analysis.
Thank you for this suggestion; on the outset we had initially considered performing this sub-analysis. However, as a result of the lack of raw data by age bracket, this was not possible. It is important to note, however, that the age range both intervention group (54.5-65.3 years) and control group (53.1-68.5 years) was similar.
- It is necessary to describe the research search and selection process in more detail.
More detail on the search and selection process has been added to the appropriate section (Page 5, Lines 202-209).
- Are there any specific reasons for not performing network meta-analysis '?
- According to Tianking et al. (BMC Medicine, 2011), a network meta-analysis, in the context of a systematic review, is a meta-analysis in which multiple treatments (that is, three or more) are compared using both direct comparisons of interventions within randomized controlled trials. They are best designed for conditions with multiple interventions, several studies with combinations of direct (or indirect) interactions and to answer relevant clinical questions by giving the “full-picture” by ranking treatments and summary outputs. The objective of our systematic review was to examine the available evidence for just two different NIBS mechanisms and explore the effectiveness of their use independently. Our objective was not to compare the effectiveness with one another. Further, there was a relative dearth of literature in this area and the outcome measures were highly heterogenous, crossing over multiple aspects of cognition. Given these considerations and limitations, our research question would not have benefited from a network meta-analysis.
- Does "Additional records identified through other sources (n = 0)" in Figure 1 mean the gray literature (ex. government reports, technical Report, thesis)? If necessary, the authors suggest that the results of searches in the gray literature be described in text.
Thank you for highlighting this – we have removed "Additional records identified through other sources (n = 0)" from the figure.
- Line 203-217: The limitations of the study need to be described in more detail.
Additional discussion has been added to the limitations section (Page 18, Lines 203-227).
Best regards,
Amer M. Burhan (senior corresponding author) on behalf of co-authors
Round 2
Reviewer 2 Report
The author gave a faithful answer to the question. Also, the contents have been revised. I suggest adding the following answer as a limitation of the study in the'Discussion section.
Subject's age is too broad in the study's inclusion criteria (18-85 years old). Age is an important influencing factor in rehabilitation. Therefore, the authors need to further analyze the effects of NIBS according to age (e.g. young, old vs. old) using meta sub-analysis.
- Thank you for this suggestion; on the outset we had initially considered performing this sub-analysis. However, as a result of the lack of raw data by age bracket, this was not possible. It is important to note, however, that the age range both intervention group (54.5-65.3 years) and control group (53.1-68.5 years) was similar.
Author Response
Thank you for this suggestion. We have added text to the limitations section on the inability to perform a sub-analysis by age due to difficulties in extracting raw data from each study please see page 18, lines 221-225
